# Investigation of Heat and Moisture Transport in Bananas during Microwave Heating Process

**Wisara Thuto and Kittichai Banjong \***

Faculty of Agro-Industry, King Mongkut's Institute of Technology Ladkrabang, Bangkok 10520, Thailand
* Correspondence: kittichai.ba@kmitl.ac.th

**Abstract:** The numerical method was used to investigate heat and moisture transport during dehydration of bananas from microwave heating. COMSOL multi-physics software was employed to perform the simulation task. A banana is defined as a porous medium. It has constituents of water, vapor, air as the liquid phase and a solid porous matrix. The numerical results of this study were validated with experimental data. The profiles of moisture, vapor and pressure are discussed in this study. Moreover, the effects of the ripening stages of the banana are examined. A higher heat flux was observed from the beginning period along with the increasing time steps until 50 s. Heat generation decreased during 50 s to 60 s, coinciding with a small rise in temperature, but the temperature gradient remained constant. The temperature distribution of both unripe and ripe banana samples was non-uniform. At the center of the banana, the temperature increased rapidly and reached its highest temperature with the negative temperature gradient toward the boundary surface. More heat generation was observed around the center region of the banana. This was due to higher moisture in comparison with the boundary surface. Heat and moisture were transported from the center of the banana to its surface. The water convective flux peaked around 11 mm from the center. The vapor pressure peaked at the center for all cases. Less heat generation within unripe bananas was observed due to the lower moisture content.

**Keywords:** banana; ripening stages; multiphase porous media; dehydration; microwave heating

---

## 1. Introduction

Bananas are a climacteric fruit grown commercially in many tropical countries [1,2]. Bananas are highly nutritious, and ripening is an indispensable process for the development of the nutritional aspects of a banana. The physical and chemical properties of bananas change at the different stages of ripening. When a banana starts to ripen, carbohydrates are initially present in the form of starches but then they transform into sugars, which are major soluble solids. Therefore, unripe bananas taste starchy and less sweet, while ripe bananas taste sweet [3,4]. Bananas are well suited to industrial processing due to their rich content of soluble solids and minerals and their low acidity [5]. Bananas can be consumed raw, cooked or processed into many types of food products [6]. In addition to preserving bananas, dehydration through many different processes has been reported by researchers (such as solar drying, convective air drying, osmotic dehydration, frying, vacuum drying, freeze drying, etc.), indicating the importance of this widely-appreciated tropical fruit [7].

Dehydration is a common method and a widely used means of food preservation, since the removal of moisture prevents the growth and reproduction of micro-organisms, enzymatic activity, and chemical reactions, thereby extending the shelf life of the food and reducing the transportation and storage costs [2,7,8]. Among the numerous dehydration methods available, microwave heating has gained popularity as an alternative dehydration technique. Microwave heating is known for its benefits, such as its ability to achieve high heating rates, reduction in treatment time, and minimal changes to the

flavor, color, and nutritional qualities of the food. There are also advantages in terms of environmental impact; microwave heating saves more energy when compared with conventional heating methods, and therefore has a reduced impact on the environment [9–11]. When food is heated in a microwave, its heat is able to reach the center of the food, whereupon the electromagnetic radiation is absorbed by dipolar molecules which occur in fat or water, thereby allowing heat to be generated. Kinetic energy is generated by the microwave radiation, causing the vibration of water molecules, which in turn creates friction. This heats the water and leads to evaporation. As a consequence, significant vapor pressure differences arise between the surface of the food and its core, which allows the moisture to be drawn out of the food item [10,12]. Microwave heating characteristics are affected by many factors; a review of microwave food processing was published showing that many interacting factors influence the temperature distribution in heated food during the microwave heating process, such as dielectric properties, physical properties, and thermal properties of the foods and heating conditions [10,13–15]. The complexity of the influence of these factors on temperature distribution during microwave heating presents a serious problem [14]; these factors affect microwave heating uniformity and can thus produce a non-uniform temperature distribution within the food [15]. Non-uniform temperature distribution is the source of many problems [13,16], such as the creation of hot spots and cold spots [17], which can seriously affect the quality of the food [9,17]. Around the location of hot spots, the temperature may be too high and can damage the nutrient substance or can cause burns or quality degradation of the final products [18]. Meanwhile, around the cold spots as well as undercooked areas, the low temperature may fail to reach the necessary level to eliminate microorganisms, which raises the risk of food deterioration and foodborne illnesses [13,18].

Experimental measurements of temperatures in a microwave oven during heating are limited due to the interference of measuring devices and the continuous nature of the process [19], which causes the temperature distribution in food materials to be very difficult to monitor and predict. This lack of information makes it difficult to determine the optimal process and to control the safety of food products. Computer simulation has been suggested as one of the best ways to understand complex microwave heating [13,20,21]; its process consists of creating geometry, forming the computational geometry with a mesh, and then solving the governing equations with a suitable numerical method [22], such as the finite difference method (FDM), finite volume method (FVM), finite element method (FEM) or boundary element method (BEM), etc. In order to simulate the microwave heating process, various researchers have used the finite element method (FEM) to examine the microwave heating process in food due to its characteristic ability to handle irregular shapes and material properties [23,24]. The ability to use a simulation eliminates the need to perform multiple experiments. There is no need to replicate conditions, and the process is sped up immeasurably. This allows research to be carried out quickly and at minimal cost [25,26]. However, the difficult stage in the simulation is the use of independent numerical codes that are highly complex in geometry generation, equation solving, and results visualization. Accordingly, programming and numerical analysis skills are required [19].

While numerous studies have been carried out involving computer simulations to replicate different food types and dehydration techniques [27–29], research to date has not provided significant insights into the processes involved from a physical perspective, and current approaches have not provided effective predictions. Among these studies, it is assumed that energy involves conductive heat transfer, while moisture involves diffusive transport, but an accurate understanding of the mechanisms underpinning the transport process when microwaves are used for heating has not yet been achieved [15].

However, in recent years, the idea of multiphase porous media has gained the interest of researchers; many researchers such as Ni et al. [30], Rakesh et al. [31], Kumar et al. [32], Ratanadecho et al. [33], and Zhang et al. [34] have devoted their research to multiphase porous medium models of microwave drying. The material sample is classified as a porous medium, while the pores duly contain the three transportable phases of water vapor, air, and water in liquid form [32,35,36]. To resolve this lack of understanding, it is necessary to develop a model which can couple heat and mass transfer to

make predictions for the distribution of moisture and the temperature within the material undergoing dehydration [32].

Although there have been numerous studies investigating the characteristics of bananas affected by different drying methods [7,37–40], there has been no studies which define the banana as a multiphase porous medium to examine the behavior of transportable phases during microwave heating. Understanding these various forms of behavior could be beneficial for the improvement, development, and design of optimum processes for dehydration in bananas with a microwave or microwave-assisted heating.

In this study, a computer simulation was used to study the dehydration phenomenon of a banana sample, at two different ripening stages, during microwave heating. The specific objectives of this study were to: (1) simulate the microwave dehydration profiles of the sample as multiphase porous media during heating and validate the results with experimental data; (2) investigate the temperature distribution of the sample; (3) investigate the transport behavior of liquid water, air, and water vapor in the sample, and (4) investigate and compare the simulation results of unripe and ripe bananas.

## 2. Materials and Methods

### 2.1. Assumptions

In order to avoid numerical errors, some assumptions were made to simplify the problem. There were seven main assumptions made in the formulation of the model in this study:

- The banana sample is considered as a porous medium, and the pores are filled with three transportable phases: liquid water, air, and water vapor.
- Local thermodynamic equilibrium exists. Thus, the solid, liquid, and gas phases are at the same average temperature at any moment in the control volume.
- Solid, liquid, and gas phases are continuous.
- The binary gas mixture of air and vapor obeys the ideal gas law.
- The non-equilibrium condition prevails during evaporation.
- Evaporation takes place throughout the entire domain.
- The shrinkage of the sample is neglected.

### 2.2. Governing Equations

#### 2.2.1. Electromagnetic Field and Heat Generation

For solving the heat distribution in the material, two kinds of mathematical models, namely, the electromagnetic field analysis model (EM model) and the heat transfer analysis model (HT model), were associated together. The internal heat generation of the material, which was used in the HT model to predict the temperature distribution, was obtained by solving the EM model [23].

The electromagnetic field distribution inside the microwave oven for heating food material is governed by Maxwell's equations [31]:

$$\nabla \times E = -j\omega\mu H \tag{1}$$

$$\nabla \times H = j\omega\varepsilon_0\varepsilon E \tag{2}$$

$$\nabla \cdot E = 0 \tag{3}$$

$$\nabla \cdot H = 0 \tag{4}$$

where $E$ is the electric field intensity (V/m); $H$ is the magnetic field intensity (A/m); $\omega$ is the angular frequency (rad/s); $\varepsilon_0$ is the permittivity of free space ($8.854 \times 10^{-12}$ F/m); $j$ is the complex number operator; and $\varepsilon$ is the complex relative permittivity, which is defined as

$$\varepsilon = \varepsilon' - j\varepsilon'' \tag{5}$$

where $\varepsilon'$ is the dielectric constant, and $\varepsilon''$ is the dielectric loss factor.

Based on Maxwell's equations, the following equation is solved to determine the electric field distribution inside a microwave oven cavity [15,41]

$$\nabla \times \left( \mu_r^{-1} \nabla \times E \right) - k_0^2 \left( \varepsilon_r - \frac{j\sigma}{\omega \varepsilon_0} \right) E = 0 \tag{6}$$

where $\mu_r$ is the relative permeability of the food material; $\varepsilon_r$ is the relative permittivity; $k_0$ is the wave number; and $\sigma$ is the electrical conductivity (S/m).

The equation below shows the complex of factors for the conversion of electromagnetic energy into thermal energy [15]

$$Q_m = \pi f \varepsilon_0 \varepsilon'' E^2 \tag{7}$$

where $Q_m$ is the heat generation due to microwaves (W/m$^3$); $f$ is the frequency (Hz); and $\varepsilon''$ is the dielectric loss factor.

### 2.2.2. Multiphase Porous Media Transport Model

In this study, a banana sample was considered as a porous medium, and the pores were filled with three transportable phases: liquid water, air, and water vapor [32,42]. The heat and mass transfer phenomena in the sample and also the phase changes in the porous media during microwave heating are shown in Figure 1. The multiphase heat and mass transport model with an additional heat generation term, calculated from the electromagnetic field distribution of the porous media, can be solved by the governing equations: mass and momentum balance equations, mass balance equations for the gas phase, energy balance equations, and evaporation rate. In this study, these governing equations are explained in detail as in Ni et al. [30] and Kumar et al. [32].

- Mass and momentum balance equations

The representative elementary volume is the sum of the volume of three phases: gas, liquid (water), and solid; thus,

$$\Delta V = \Delta V_g + \Delta V_w + \Delta V_s \tag{8}$$

where $\Delta V$ is the representative elementary volume (m$^3$); $\Delta V_g$ is the volume of gas (m$^3$); $\Delta V_w$ is the volume of water ($m^3$), and $\Delta V_s$ is the volume of solid (m$^3$).

Equivalent porosity is defined as

$$\varphi = \frac{\Delta V_g + \Delta V_w}{\Delta V} \tag{9}$$

where $\varphi$ is the equivalent porosity.

Equivalent saturations of liquid and gas are defined as

$$S_w = \frac{\Delta V_w}{\Delta V_w + \Delta V_s} = \frac{\Delta V_w}{\varphi \Delta V} \tag{10}$$

and

$$S_g = \frac{\Delta V_g}{\Delta V_w + \Delta V_g} = \frac{\Delta V_g}{\varphi \Delta V} = 1 - S_w \tag{11}$$

where $S_w$ is the equivalent saturation of water, and $S_g$ is the equivalent saturation of gas.

The mass densities of vapor, air, and their mixture are given by

$$\rho_v = \frac{\Delta m_v}{\Delta V_g} = \frac{p_v M_v}{RT} \tag{12}$$

$$\rho_a = \frac{\Delta m_a}{\Delta V_g} = \frac{p_a M_a}{RT} \tag{13}$$

$$\rho_g = \frac{\Delta m_v + \Delta m_a}{\Delta V_g} = \frac{P M_g}{RT} = \rho_v + \rho_a \tag{14}$$

where $\rho_v$ is the mass density of vapor (kg/m$^3$); $\rho_a$ is the mass density of air (kg/m$^3$); $\rho_g$ is the mass density of gas (kg/m$^3$); $p_v$ is the partial pressure of vapor (Pa); $p_a$ is the partial pressure of air (Pa); $P$ is the total pressure (Pa); $M_v$ is the molar mass of vapor (kg/mol); $M_a$ is the molar mass of air (kg/mol); $M_g$ is the molar mass of gas (kg/mol); $\Delta m_v$ is the mass of vapor in a representative elementary volume (kg); $\Delta m_a$ is the mass of air in a representative elementary volume (kg); $R$ is the universal gas constant (J/mol K), and $T$ is the temperature of the material (K).

For the gas mixture of air and vapor, the total pressure is given by

$$P = p_v + p_a \tag{15}$$

The mass concentrations of vapor, air, and liquid water are given by

$$c_v = \frac{\Delta m_v}{\Delta V} = \frac{p_v M_v \varphi S_g}{RT} \tag{16}$$

$$c_a = \frac{\Delta m_a}{\Delta V} = \frac{p_a M_a \varphi S_g}{RT} \tag{17}$$

$$c_w = \frac{\Delta m_w}{\Delta V} = \frac{\rho_w \Delta V_w}{\Delta V} = \rho_w \varphi S_w \tag{18}$$

where $c_v$ is the mass concentration of vapor (kg/m$^3$); $c_a$ is the mass concentration of air (kg/m$^3$); $c_w$ is the mass concentration of liquid water (kg/m$^3$); $\Delta m_w$ is the mass of liquid water in a representative elementary volume (kg); and $\rho_w$ is the density of water (kg/m$^3$).

The mass conservation equation for liquid water considers gas pressure driven flow, capillary diffusion, and evaporation of liquid water to vapor. The equations for the mass concentration of liquid water can be written as

$$\frac{\partial}{\partial t}(\varphi S_w \rho_w) + \nabla\cdot\left(-\rho_w \frac{k_w k_{r,w}}{\mu_w} \nabla P - D_c \nabla c_w\right) = -R_{evap} \tag{19}$$

where $k_w$ is the intrinsic permeability of water (m$^2$); $k_{r,w}$ is the relative permeability of water; $\mu_w$ is the viscosity of water (Pa s); $D_c$ is the capillary diffusivity (m$^2$/s); and $R_{evap}$ is the evaporation rate of liquid water to water vapor (kg/m$^3$s).

The mass balance equation for the vapor component of the gas phase includes bulk flow, binary diffusion, and phase change, and is given by

$$\frac{\partial}{\partial t}\left(\varphi S_g \rho_g \omega_v\right) + \nabla\cdot\left(-\rho_g \omega_v \frac{k_g k_{r,g}}{\mu_g} \nabla P - \varphi S_g \rho_g D_{eff,g} \nabla \omega_v\right) = R_{evap} \tag{20}$$

where $\omega_v$ is the mass fraction of vapor; $k_g$ is the intrinsic permeability of gas (m$^2$); $k_{r,g}$ is the relative permeability of gas; $\mu_g$ is the viscosity of gas (Pa s); and $D_{eff,g}$ is the binary diffusivity of vapor and air (m$^2$/s).

The mass fraction of air can be calculated from the expression

$$\omega_a = 1 - \omega_v \tag{21}$$

where $\omega_a$ is the mass fraction of air.

- Mass balance equations for the gas phase

The overcall mass balance for the gas phase is defined as

$$\frac{\partial}{\partial t}\left(\varphi S_g \rho_g\right) + \nabla\cdot\left(-\rho_g \frac{k_g k_{r,g}}{\mu_g}\nabla P\right) = R_{evap} \tag{22}$$

- Energy balance equation

The energy balance equation is described by the following equation:

$$\rho_{eff}c_{peff}\frac{\partial T}{\partial t} + \nabla\cdot\left(\vec{n}_g h_g + \vec{n}_w h_w\right) = \nabla\cdot\left(k_{eff}\nabla T\right) - h_{fg}R_{evap} + Q_m \tag{23}$$

where $\vec{n}_w$ is the water flux (kg/m$^2$ s); $\vec{n}_g$ is the gas flux (kg/m$^2$ s); $T$ is the temperature of each phase (K); $h_g$ is the enthalpy of gas (J); $h_w$ is the enthalpy of water (J); $h_{fg}$ is the latent heat of evaporation (J/kg); $\rho_{eff}$ is the effective density (kg/m$^3$); $c_{peff}$ is the effective specific heat (J/kg K); and $k_{eff}$ is the effective thermal conductivity (W/m K).

The flux of water and gas can be calculated from the expression:

$$\vec{n}_w = -\rho_w \frac{k_w k_{r,w}}{\mu_w}\nabla P - D_c \nabla c_w \tag{24}$$

and

$$\vec{n}_g = -\rho_g \omega_v \frac{k_g k_{r,g}}{\mu_g}\nabla P - \varphi S_g \rho_g D_{eff,g}\nabla\omega_v \tag{25}$$

The effective properties taking mass and volume changes are described by following equations:

$$\rho_{eff} = \varphi\left(S_g\rho_g + S_w\rho_w\right) + (1-\varphi)\rho_s \tag{26}$$

$$c_{peff} = m_g\left(\omega_g c_{pg} + \omega_a c_{pa}\right) + m_w c_{pw} + m_s c_{ps} \tag{27}$$

$$k_{eff} = \varphi\left(S_g k_{th,g} + S_w k_{th,w}\right) + (1-\varphi)k_{th,s} \tag{28}$$

where $\rho_s$ is the solid density (kg/m$^3$); $c_{pg}$, $c_{pw}$, and $c_{ps}$ are the specific heat capacities of gas, water, and solid (J/kg K), respectively; $k_{th,g}$, $k_{th,w}$, and $k_{th,s}$ are the thermal conductivities of gas, water, and solid (W/m K), respectively; and $m_g$, $m_w$, and $m_s$ are the mass fractions of gas, water, and solid (kg), respectively.

- Evaporation rate

The evaporation rate can be calculated from the expression:

$$R_{evap} = K_{evap}\frac{M_v}{RT}\left(p_{v,eq} - p_v\right) \tag{29}$$

where $K_{evap}$ is the evaporation constant (1/s), and $p_{v,eq}$ is the equilibrium vapor pressure (Pa).

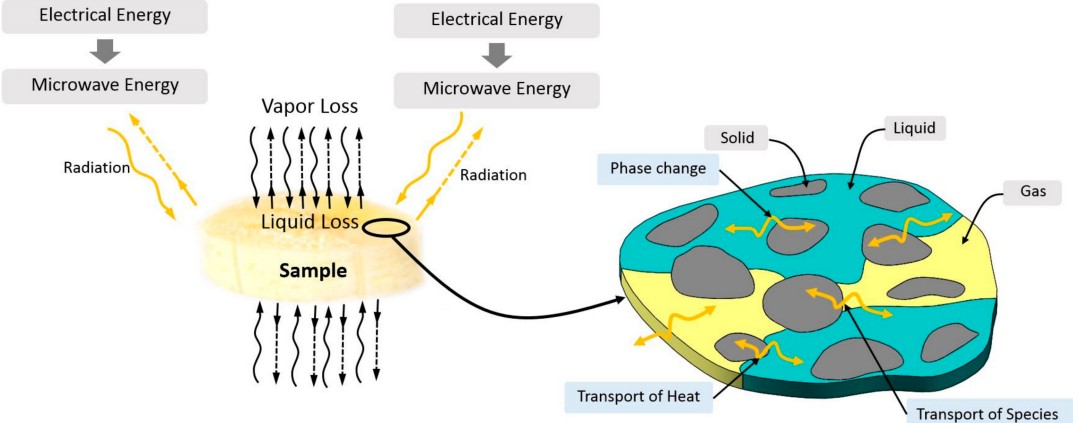

**Figure 1.** Heat and mass transfer of a sample with representative elementary volume (REV) showing transport, reaction, and phase change in porous media.

*2.3. Geometry*

The geometric model of the microwave system used in this study was developed based on the real structure and size of the domestic 800 W microwave oven that was used in the experimental procedures. It was composed of an oven cavity, waveguide, and turntable (glass plate), as shown in Figure 2, and the dimensions are given in Table 1. The excitation for the microwave oven is through a rectangular waveguide. At an excitation frequency of 2.45 GHz, the $TE_{10}$ mode is the only propagation mode [41]. The geometry of the banana samples was generated into a transversal shape, as shown in Figure 3, and the dimensions of banana samples are given in Table 1.

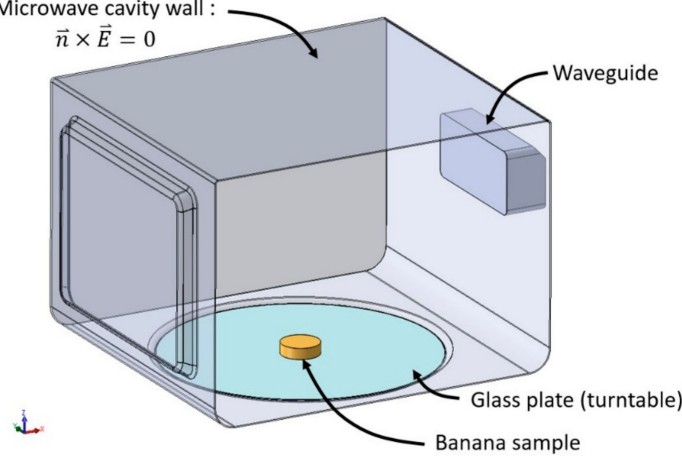

**Figure 2.** The geometrical model.

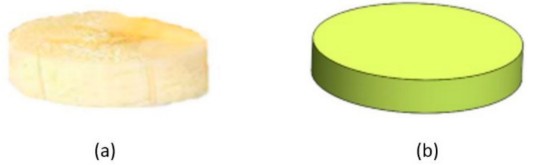

**Figure 3.** The sample geometry. (**a**) Transversal banana slice, and (**b**) geometry of banana slice used for simulation.

**Table 1.** Summary of input parameters used in the simulations.

| Parameter | Value | Unit | Source |
|---|---|---|---|
| Heating time | 60 | s | In this study |
| Microwave frequency | 2450 | MHz | [43] |
| Oven cavity dimensions (w × d × h) | 319 × 336 × 204 | mm | [43] |
| Waveguide dimensions (w × d × h) | 50 × 78 × 18 | mm | [43] |
| Turntable diameter | 272 | mm | [43] |
| Initial air inside cavity temperature | 25 | °C | In this study |
| Initial banana temperature | 25 | °C | In this study |
| Banana dimension (D × h) | 30 × 5 | mm | In this study |
| Initial moisture content | | | |
| - Unripe banana | 45 | %wb | In this study |
| - Ripe banana | 65 | %wb | In this study |
| Specific heat | | | |
| - Unripe banana | 2430 | J/kg K | [44] |
| - Ripe banana | 3430 | J/kg K | [44] |
| - Water | 4187 | J/kg K | [32,45] |
| - Vapor | 1900 | J/kg K | [32,45] |
| - Air | 1005.683 | J/kg K | [32,45] |
| Density | | | |
| - Unripe banana | 1320 | kg/m$^3$ | [44] |
| - Ripe banana | 870 | kg/m$^3$ | [44] |
| - Water | 1000 | kg/m$^3$ | [32,46] |
| - Vapor | Ideal gas law | kg/m$^3$ | [31,32] |
| - Air | Ideal gas law | kg/m$^3$ | [31,32] |
| Thermal conductivity | | | |
| - Unripe banana | 0.41 | W/m K | [44] |
| - Ripe banana | 0.97 | W/m K | [44] |
| - Water | 0.644 | W/m K | [31,32,46] |
| - Gas | 0.026 | W/m K | [31,32,46] |
| Initial Equivalent porosity | | | |
| - Unripe banana | 0.75 | - | [30] |
| - Ripe banana | 0.83 | - | [47] |
| Initial water saturation | | | |
| - Unripe banana | 0.5 | - | [30] |
| - Ripe banana | 0.8 | - | [47] |
| Intrinsic permeability | | | |
| - Water | $5 \times 10^{-14}$ | m$^2$ | [30,31] |
| - Vapor and air | $10 \times 10^{-14}$ | m$^2$ | [30,31] |
| Dielectric constant | 60 | - | [48] |
| Dielectric loss | 18 | - | [48] |
| Heat transfer coefficient | 20 | W/m$^2$ K | [30] |
| Mass transfer coefficient | 0.01 | m/s | [30] |
| Binary diffusivity | $2 \times 10^{-6}$ | m$^2$/s | [31,32] |
| Latent heat of evaporation | $2.26 \times 10^6$ | J/kg | [31,32,49] |
| Molecular weight of gas | 28.966 | g/mol | [32,49] |
| Molecular weight of vapor | 18.016 | g/mol | [32,49] |
| Molecular weight of water | 18.016 | g/mol | [32,49] |
| Universal gas constant | 8.314 | J/mol K | [32,49] |
| Ambient pressure | 101325 | Pa | [31,32,49] |

### 2.4. Mesh Qualities

To achieve a good prediction accuracy and reduced computational time, the geometric models require an appropriate mesh size for the domains. Tetrahedral elements were assigned to the domains, including the oven cavity, waveguide, turntable, banana sample and air in the oven cavity. The main research objective of this paper is to investigate the heat distribution of the banana; therefore, extremely fine mesh refinement levels were enforced for the banana sample, while fine mesh refinement levels were selected for the other domains. The computational domain for the heat distribution was used for the banana sample only. Figure 4 shows the meshing scheme of the microwave oven and the banana sample.

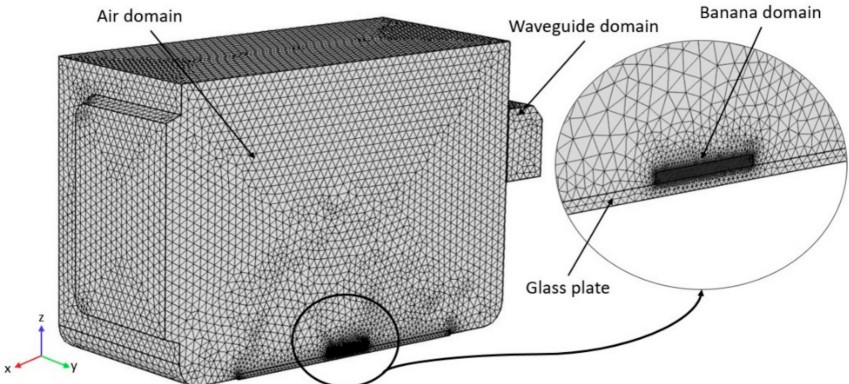

**Figure 4.** Meshing scheme of the materials.

### 2.5. Boundary Conditions

In a microwave oven cavity, the walls of the cavity are considered to be perfect electrical conductors, where the following boundary condition applies

$$E_{tangential} = 0 \tag{30}$$

The boundary conditions for heat and mass transfer at the transport boundaries of the food sample for this study are similar in their detail to those of Rakesh et al. [31] and Kumar et al. [32].

The heat and mass transfer that take place at the transport boundaries are described by the following equations

$$\vec{n}_{w,s} = h_{mv}\varphi S_w \frac{(p_v - p_{v\,air})}{RT} \tag{31}$$

and

$$\vec{n}_{v,s} = h_{mv}\varphi S_g \frac{(p_v - p_{v\,air})}{RT} \tag{32}$$

where $\vec{n}_{w,s}$ is the total water flux at the surface (kg/m$^2$ s); $\vec{n}_{v,s}$ is the total vapor flux at the surface (kg/m$^2$ s); $h_{mv}$ is the mass transfer coefficient (m/s); and $p_{v,air}$ is the vapor pressure of ambient air (Pa).

The pressure was set to ambient at all surfaces of the sample. The boundary condition can be expressed as

$$P = P_{amb} \tag{33}$$

where $P_{amb}$ is the ambient pressure (Pa).

The loss of heat due to the evaporation of water and the removal of liquid water and vapor is also included in the boundary condition for heat transfer, and it can be expressed as

$$q_{surf} = h_T(T - T_{air}) - h_{mv}\varphi S_w \frac{(p_v - p_{v,air})}{RT} h_{fg} - h_{mv}\varphi S_g \frac{(p_v - p_{v,air})}{RT} C_{p,v}T \tag{34}$$

where $h_T$ is the heat transfer coefficient (W/m$^2$K); $T_{air}$ is the drying air temperature (K); and $C_{p,v}$ is the specific heat of vapor at a constant volume (J/kg K).

### 2.6. Input Parameters and Initial Conditions

In this study, some input parameters and initial conditions were taken from the literature and summarized in Table 1.

With respect to the stages of ripeness, there are many changes in the characteristics of the banana that might affect the heat distribution during heating. For example, the moisture content and total soluble solids have mainly independent effects on dielectric properties. Huang [4] reported that the values of both the dielectric constant ($\varepsilon'$) and the dielectric loss ($\varepsilon''$) of the banana remained stable during different stages of ripeness, because the selected conditions were limited to room temperature, and the microwave frequency ranges were restricted, hence discrimination related to frequency was not possible. Therefore, in the present study, which has similar conditions to Huang's study [4], these parameters were considered as constant.

### 2.7. Numerical Solution

In our study, a workstation based on the Intel Xeon E7 and 32 GB of RAM with COMSOL Multiphysics software was selected for solving the mathematical models by use of the finite element method (FEM). Finite elements are analyzed in the simulations by the mesh in the domains, utilizing numerical solvers [8]. The coupling of physics and the solution process is shown in Figure 5.

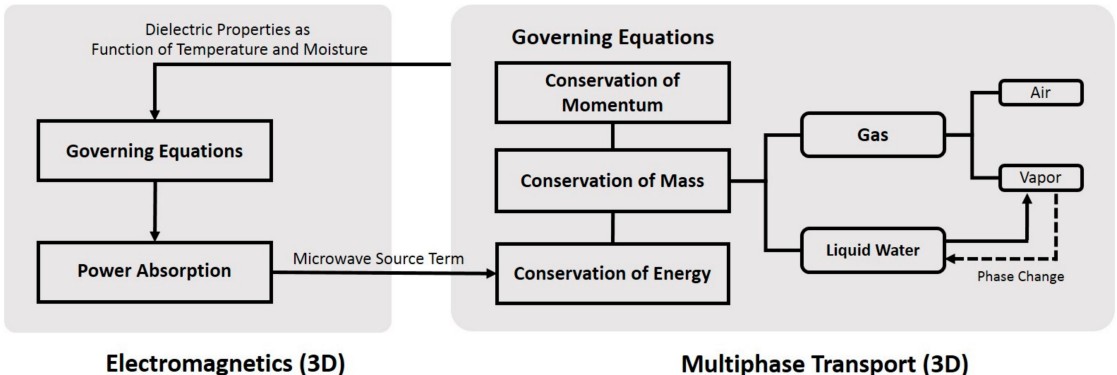

**Figure 5.** Flow chart of the sequence of steps followed to develop the computational model.

### 2.8. Experiment Procedures

Bananas (*Musa sapientum* Linn.) that had similar characteristics with no evidence of mechanical damage were purchased from a local market. The bananas were classified according to a peel color index, CSIRO [50]. Samples at different ripening stages (e.g., unripe bananas (stage 1) and ripe bananas (stage 6)), were selected as samples for this study. The initial moisture contents of the unripe and ripe bananas were estimated to be approximately 45% wb and 65% wb, respectively. The banana samples from each ripening stage were peeled and sliced into a transversal shape, with dimensions as shown in Table 1. A sample was placed on the center of the turntable plate inside the microwave oven cavity. Heating was performed in a microwave oven using the maximum level at 800 W for 60 s, and the dehydration temperatures were monitored and recorded using a thermal imaging camera (FLIR E40, FLIR Systems, Inc., USA, 320 × 240 pixels, thermal sensitivity/NETD <0.07 °C) at an interval of 10 s. The microwave experiment setup scheme is shown in Figure 6. Each experiment was performed in triplicate in order to obtain average temperature data.

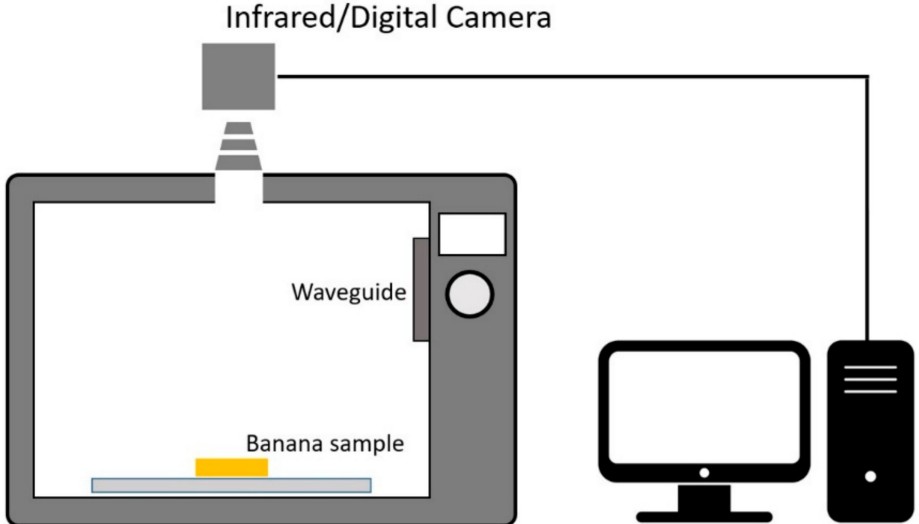

**Figure 6.** Schematic diagram of microwave heating experimental.

Moisture loss was determined through weighing the sample during the experimentation; the sample was taken out from the microwave oven cavity, weighed on a digital balance, with weights recorded at every 10 s, and placed back into the cavity immediately after weighing. The dehydration procedure was repeated, and the dehydration experiment was stopped at 60 s of heating time.

*2.9. Model Validation*

To validate the simulation results, the simulated average moisture content and temperature profiles were compared with the data collected from the experiment. The accuracy of the simulation results was assessed by determining the coefficient of determination (R-squared; $R^2$). Note that the maximum value of R-squared is 1.0, which indicates a perfect prediction. In general, a value above 0.75 normally indicates a useable result [51].

## 3. Results and Discussion

*3.1. Experimental Validation of Moisture Content*

During heating, the moisture content continually decreased due to the evaporation of water. As shown in Figure 7, the simulation model showed good agreement with the experiment, in which the R-squared values were 0.9834 for unripe bananas and 0.9908 for ripe bananas. Both the experiment and the simulation model showed that the average moisture content of samples after the end of the process at 60 s decreased from 45% wb to 5% wb for unripe bananas and 65% wb to 15% wb for ripe bananas.

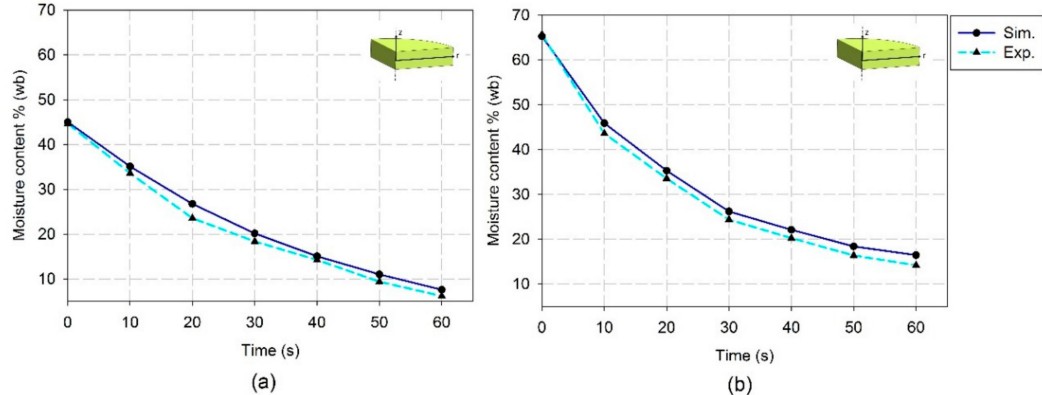

**Figure 7.** Comparison between predicted and experimental values of average moisture content of the sample during dehydration; (**a**) unripe bananas; (**b**) ripe bananas.

### 3.2. Temperature Distribution Profile

Figure 8 shows the simulated temperature profiles of the unripe and ripe bananas during microwave heating compared with the experimental results, at various distances from the center of the sample. It is apparent that the agreement between the results of the simulation and the experiments was qualitatively consistent. Additionally, this simulation can be considered sufficiently reliable to predict the temperature profiles of unripe and ripe bananas, due to the fact that the R-squared values of the simulated temperature profiles were calculated as 0.9778 and 0.9804, respectively.

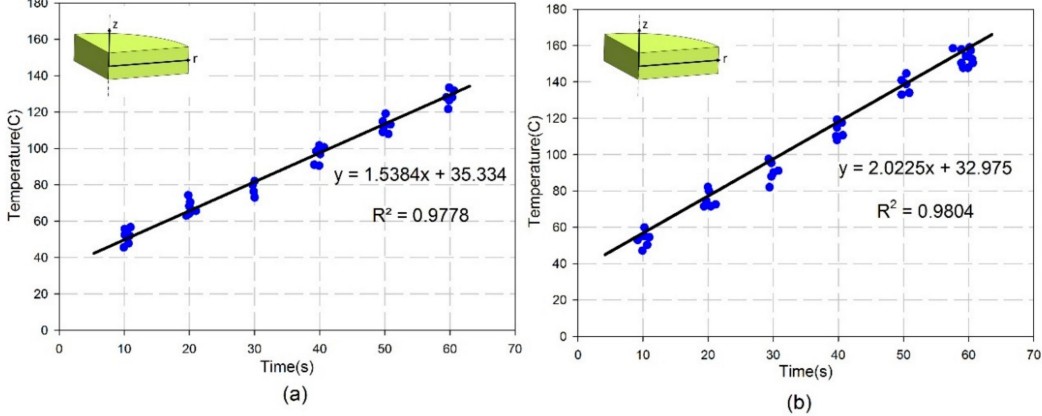

**Figure 8.** The temperature profile comparison between experiment and simulation; (**a**) unripe bananas; (**b**) ripe bananas.

Considering the results as shown in Figure 9, the temperature was plotted for the distance from the left to the right edge of the sample, at intervals of 10 s. This allows the investigation of temperature evolution both externally and internally with regard to the sample. Both unripe and ripe bananas show similar patterns in their temperature profiles: the internal temperature was higher than the external temperature, and the highest temperature was found at the center of the sample.

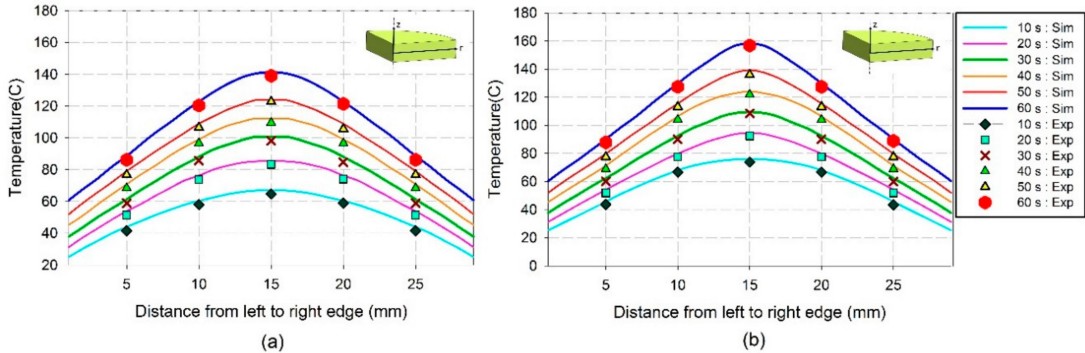

**Figure 9.** Temperature at different distances; (**a**) unripe bananas; (**b**) ripe bananas.

In comparing the unripe and ripe bananas, the range of temperatures for the unripe and ripe bananas from the start to the end of the heating process was about 25–140 °C and 25–150 °C, respectively. The highest temperatures, which were found at the centers of the samples, were about 140 °C for unripe and 150 °C for ripe bananas. A hot spot was observed at the top surface, at the center of the banana, as shown in Figure 10, due to a high moisture portion located near the surface. Therefore, high thermal energy was generated due to the highest microwave absorption. Heat flux was transported from the center toward all boundary surfaces. Higher heat flux was observed during the initial period along with the increasing time steps until 50 s, which corresponds with the increasing temperature gradient, as seen in Figure 10. Heat generation decreased from 50 s to 60 s in coincidence with little temperature raised, and the temperature gradient remained constant.

Figure 10 illustrates the comparison between experimental data and computational results for temperature distribution on the top surface of the sample. It can be seen that both the unripe and ripe bananas show similar patterns of non-uniform temperature distributions. The temperature peaked at the center, where microwaves are converted to thermal energy. The surface region of the sample, in contrast, remains cools because the air surrounding the food is able to exert a cooling influence [52]. While the internal temperature increases continually with heating time and is distributed from the center to the surface, the surface has a lower temperature, resulting from evaporative cooling [52]. After reaching its highest temperature, the temperature slowly decreases until the end of the process as a result of the moisture content being gradually removed from the sample [52]. In unripe bananas (Figure 10a), the rate of increase in temperature over time is relatively slow in comparison to that observed in ripe bananas (Figure 10b). This is because the microwaves are absorbed at a lower rate because of the lack of moisture.

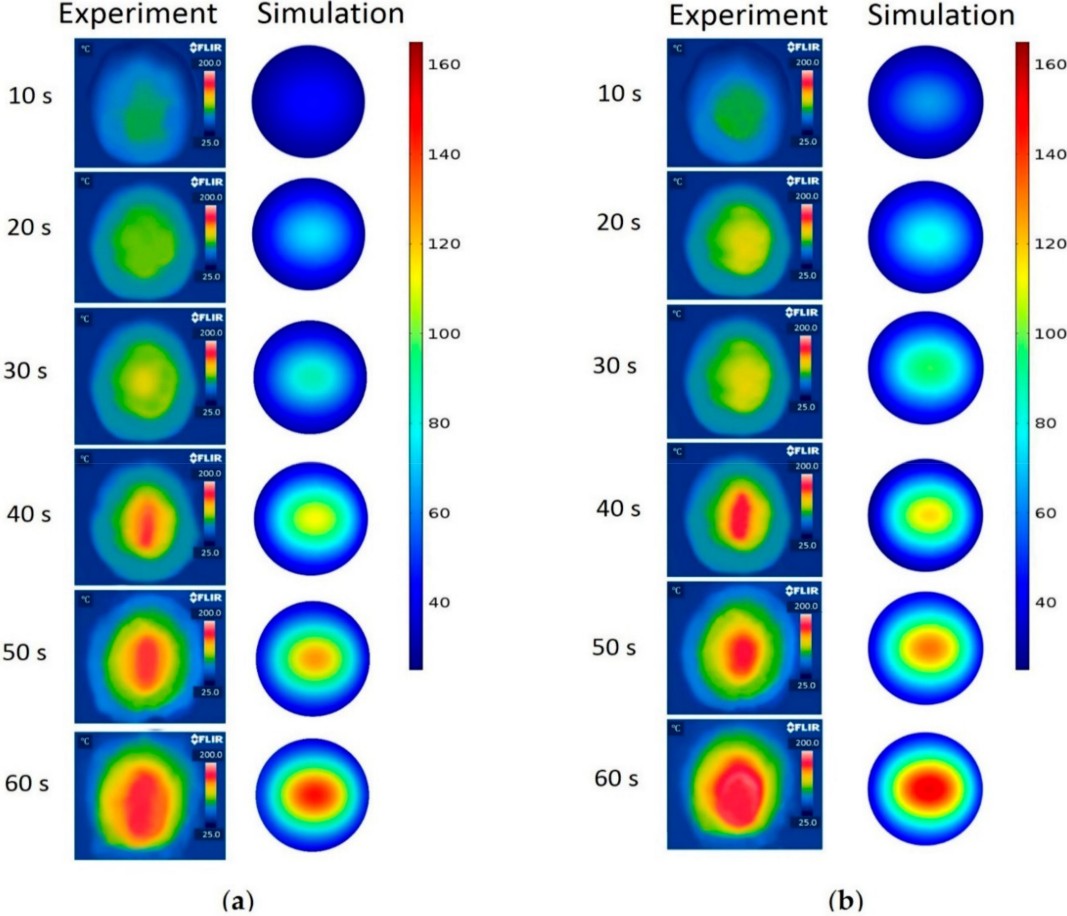

**Figure 10.** Comparison between infrared thermal image and simulation model of temperature distributions; (**a**) unripe banana; (**b**) ripe banana.

Figure 11 shows a photograph of the banana samples that were used in the experiments, at various time intervals. It is apparent that the agreement between the results of the simulation and the experiments was qualitatively consistent, particularly at the center, as the higher temperature region caused the banana sample to burn.

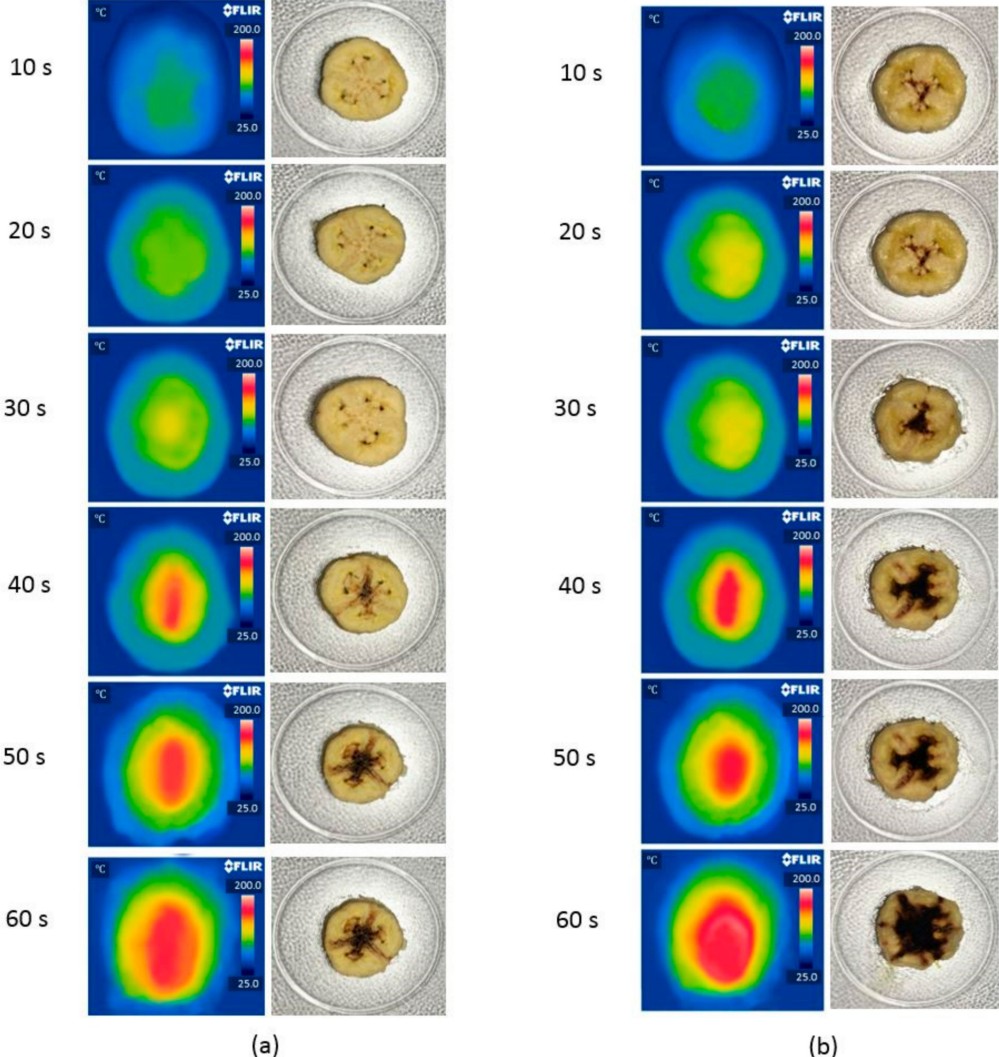

**Figure 11.** The infrared thermal images with photos of banana samples used in the experiments at different time intervals; (**a**) unripe bananas; (**b**) ripe bananas.

### 3.3. The Moisture Profile

Distance-dependent changes in the water saturation profiles of bananas during dehydration are shown in Figure 12. Both unripe and ripe bananas had similar water saturation profiles, in which there were high moisture saturation regions at the center, toward 10 mm in the radial direction; and the moisture saturation sharply decreased from 10 mm to the surface of the banana at 15 mm in the radial direction. This result indicates that a higher moisture content exists in the center than on the surface during heating, due to the fact that the liquid water that migrates from the internal part of the sample to the surface is transformed to vapor and vaporizes from the surface to the surroundings [33,53,54]; accordingly, there is a lower moisture content at the surface. In addition, it can be seen that at the central region, the water saturation at the end of the process decreased slightly due to a lower rate of evaporation, when compared with the surface region. After heating for 60 s, the water saturation at the center of the sample dropped from 0.5 to 0.45 for unripe bananas and 0.8 to 0.76 for ripe bananas. Kumar et al. [32] and Mercier et al. [55] also found this phenomenon.

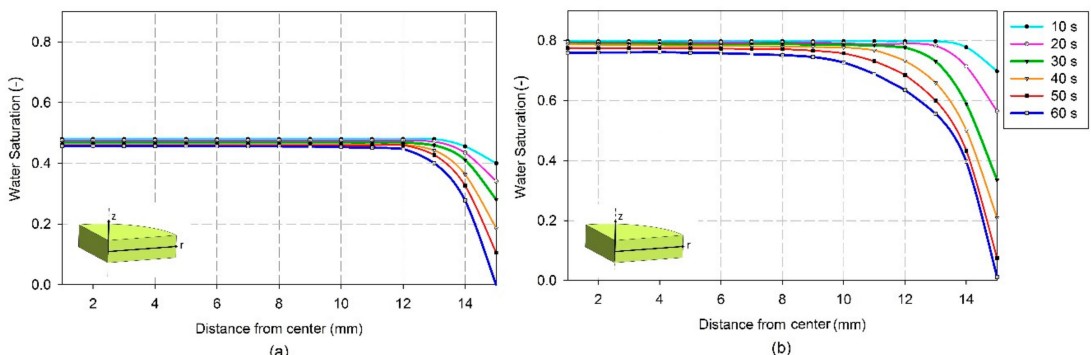

**Figure 12.** The water saturation profiles of (**a**) unripe bananas, and (**b**) ripe bananas.

### 3.4. The Vapor Mass Fraction Profile

The vapor mass fractions at various time intervals are shown in Figure 13. Both unripe and ripe bananas show similar results, in which the vapor mass fraction decreases gradually with distance from the center up to 10 mm from the center. A higher decreasing rate of mass fraction was revealed around the surface region, and a higher gradient profile developed along with the increasing time steps. The vapor mass fraction increases with time, at any distance, because of moisture evaporation in correspondence with the raising temperature. The highest vapor mass fraction occurs at the center of the sample. Vapor mass fractions correlate with the temperature profiles. The higher vapor mass fractions are found in the regions where the high temperatures are found, at any time interval during the dehydration process. This was due to the internal region having a greater rate of moisture evaporation than moisture transport to the surface region. For this reason, the lower vapor mass fractions occur at the surface due to the transportation of moisture to the surrounding or ambient air [32]. Again, due to the lower initial moisture content of unripe bananas, the vapor mass fraction is also lower than that of ripe bananas at the same time interval.

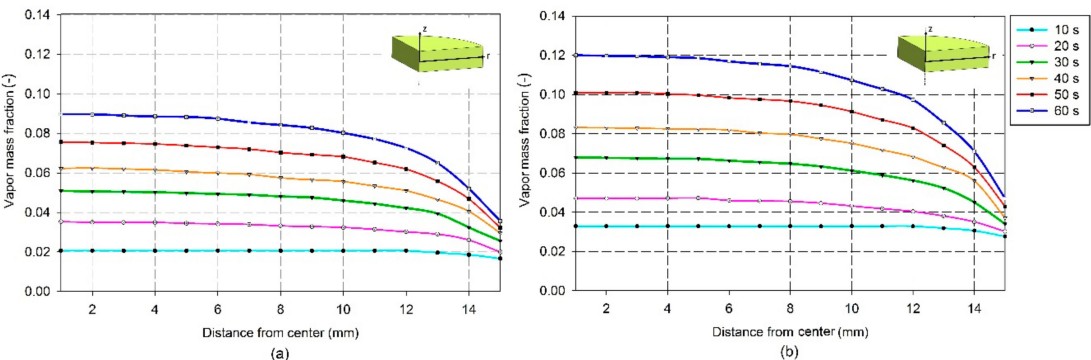

**Figure 13.** The vapor mass fractions profiles of (**a**) unripe bananas, and (**b**) ripe bananas.

### 3.5. The Vapor Pressure Profile

The vapor pressure profiles within the banana samples at each time interval are shown in Figure 14. The results show that for both unripe and ripe bananas, the vapor pressure is higher at the internal region of the sample, with the highest vapor pressure occurring at the center. The unripe bananas have lower vapor pressure levels compared with the ripe bananas at the same time interval. These behaviors of vapor pressure are similar to the moisture and vapor mass fraction profiles, which correlate with the temperature profiles.

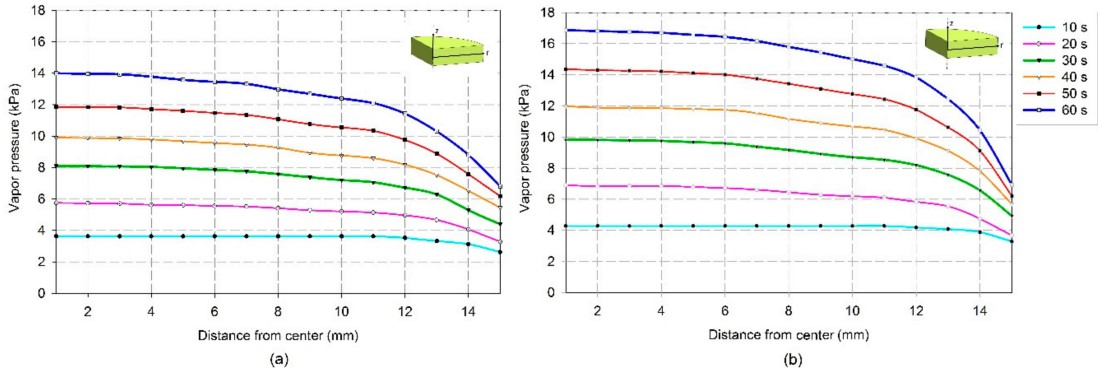

**Figure 14.** The vapor pressure profiles of (**a**) unripe bananas, and (**b**) ripe banana.

Halder et al. [56] found a similar result to our study in that at the beginning of heating, vapor pressure in the core region was lower, which was caused by the condensation of migrated water and vapor from the surface to the internal region. However, when the evaporation of moisture increased as temperature rose in the center region, then vapor saturation in the region rose, which led to an increase in vapor pressure, which pushed the vapor flow through the porous medium toward the surface [54]. In addition, on the surface of the material, where vapor pressure was close to ambient pressure, the vapor could be readily diffused into the surrounding air, allowing the surface to dry [32,57].

### 3.6. Vapor and Water Fluxes

Figures 15 and 16 show the liquid water fluxes due to the capillary diffusion and gas pressure gradient of the banana, respectively. It can be seen that both the liquid water fluxes show results with quite similar patterns, in which the liquid water fluxes decrease over time. The higher liquid water fluxes occur near the surface region, while the liquid water fluxes decrease at the internal regions towards the center, with the lowest value occurring at the center. The peaks of the fluxes are found at small distances from the surface. This could be due to the fact that at the surface, the moisture is higher, because the external region close to the surface has less moisture and hence the concentration gradient is higher, which leads to greater fluxes [32].

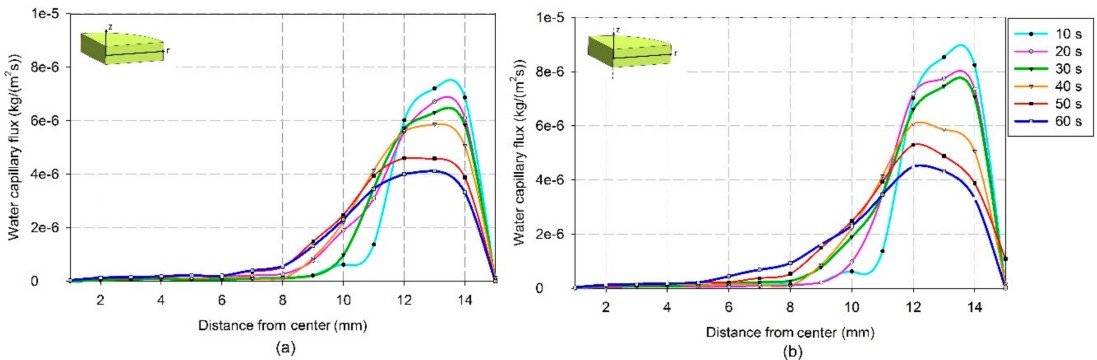

**Figure 15.** Water flux due to capillary of (**a**) unripe bananas, and (**b**) ripe bananas.

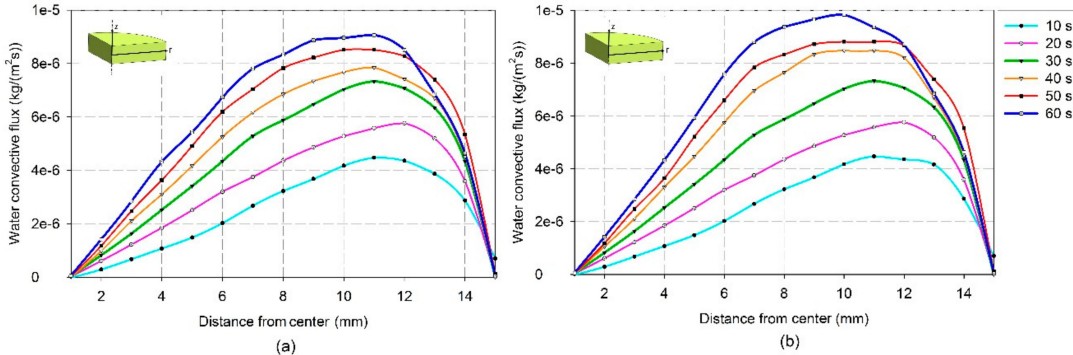

**Figure 16.** Water flux due to gas pressure of (**a**) unripe bananas, and (**b**) ripe bananas.

It can be seen that the vapor flux due to the binary diffusion of the banana (Figure 17) and the vapor flux due to gas pressure (Figure 18) have similar profile patterns. The figures show that both of these vapor fluxes increase over the heating time, and sharply increase at about 10 mm from the center, with the highest fluxes occurring at the surface. With time, it can also be seen that both of these vapor fluxes move slightly towards the internal region. However, these slight changes are insignificant in distance, and both vapor fluxes still occur at or near the surface. As a result, due to the gradient of the vapor being very high near the surface, higher diffusive flux is caused, along with more moisture loss [32].

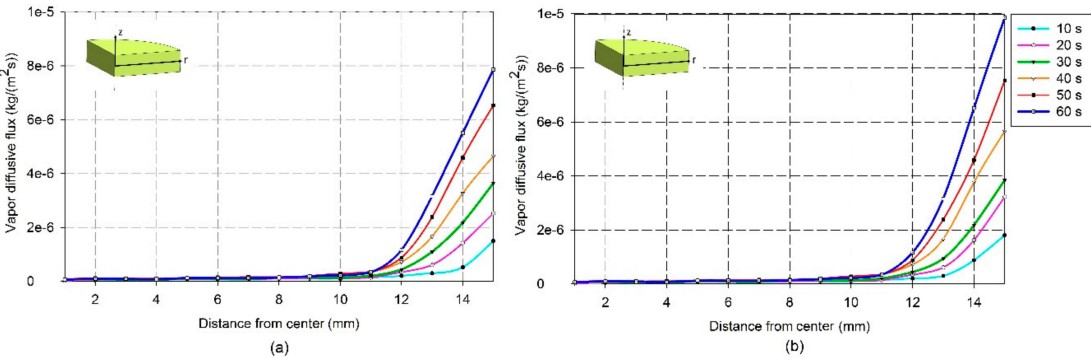

**Figure 17.** Vapor flux due to binary diffusion of (**a**) unripe bananas and, (**b**) ripe bananas.

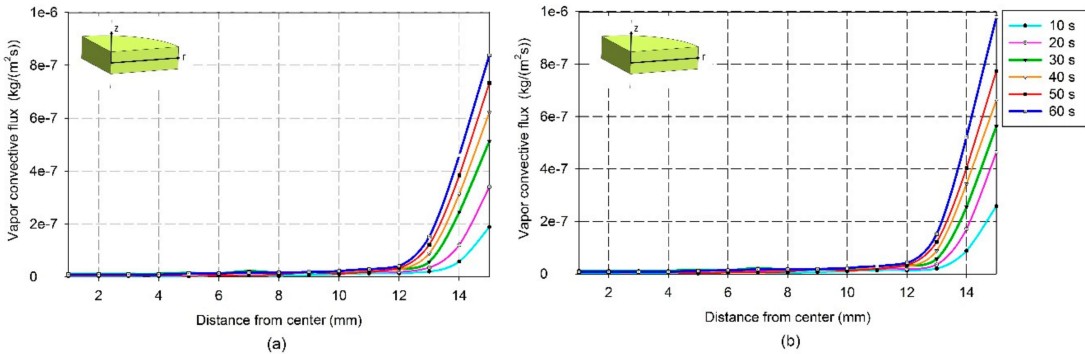

**Figure 18.** Vapor flux due to gas pressure of (**a**) unripe bananas and, (**b**) ripe bananas.

Unripe and ripe bananas show similar profile patterns in both water and vapor flux. However, due to the initial moisture content, a slight difference can be seen in the levels of fluxes at the same time interval.

### 3.7. Identification of Hot Spot

In Figure 10, the images in the same row show the temperature profiles at an interval of 10 s, in which the darkest red color indicates the highest temperature, and conversely, the darkest blue color indicates the lowest temperature. In addition, the simulated model correlated well with infrared thermal images that can identify cold and hot spots, which were located at the surface region and at the center of the sample, respectively. The average temperatures at cold and hot spots of unripe bananas were found to be 60 °C and 140 °C, respectively. Meanwhile, the average temperatures at cold and hot spots of ripe bananas corresponded to 65 °C and 150 °C, respectively.

Hot spot occurrence is influenced by the electromagnetic and thermodynamic features of the microwave system and the material [58,59]. The geometry of the material affects the heat distribution during microwave heating due to the penetration depth of the material. The penetration depth can be measured by the electromagnetic radiation that penetrates into a material. If the penetration depth is minor compared with the dimensions of the material, most of the energy is absorbed near the surface, leaving the center cold. Conversely, if the penetration depth is significant, reasonable amounts of energy reach the center and focusing occurs [60,61].

The penetration depth can be given as [61]

$$d_p = \frac{\lambda}{2\pi} \cdot \left[ \frac{2}{\varepsilon' \left[ \sqrt{1 + tan^2\delta} - 1 \right]} \right]^{1/2} \tag{35}$$

where $d_p$ is the penetration depth in the material (mm); $\lambda$ is the wavelength (m), and $\tan\delta$ is the dielectric loss tangent, which is given by [61]

$$\tan\delta = \frac{\varepsilon''}{\varepsilon'} \tag{36}$$

Calculated by Equation (35), the penetration depth of a banana sample is 8.37 mm, due to the fact that the thickness of both unripe and ripe banana samples was set as 5 mm, which is less than the penetration depth. Therefore, the results are in agreement with those mentioned above in that most of the energy reaches to the center and focusing occurs; as a result, the hot spot was generated [60,61]. Chandrasekaran et al. [10] also found that hot spots occur at the center for slab-shaped samples.

In addition, at the location of the hot spot, the temperature may be too high, which would cause overheating of the material or damage the nutrients' substance. In contrast, at the cold spot, the low temperature may fail to reach the damage temperature of the microorganisms, which would raise the risk of food deterioration [10,17]. Thus, the cold and hot spots are very important to the design of the optimum process of microwave heating.

### 3.8. Unripe Banana vs. Ripe Banana

As can be seen in Figure 10a, unripe bananas as a low moisture material had a fairly slow temperature increase, since the rate of the microwave absorption was lower due to its low moisture content. At about 20 s of heating time, the first signs of heat generation at the center of the sample could be clearly seen, and it reached 100 °C at about 30 s. As the temperature increased, the rate of evaporation also increased, and the internal pressure started to build up. Thus, the lower internal temperatures caused a lower rate of evaporation and lower pressure generation. This process created a pressure gradient, which caused a fast movement of the liquid water and water vapor towards the surface of the material.

Ripe bananas with high initial moisture content were considered as high moisture material, as under the same heating condition, the patterns of the temperature, moisture, and pressure profiles were similar to those of unripe bananas. However, ripe bananas showed a slight elevation in the level of values, as can be seen in Figure 10b. The temperature profiles of ripe bananas could reach 100 °C

at the center of the sample at about 20 s, faster than unripe bananas. With temperature dependence, the higher internal temperatures caused a higher rate of evaporation and higher pressure generation; consequently, the heating rate was also higher than that in the low moisture material of unripe bananas.

Therefore, the sorting or grading of bananas into the correct ripening stages was an important process to achieve the optimum process for the dehydration of bananas by microwave heating. This was due to the differences in initial moisture content, which could generate different microwave absorption profiles, causing a difference in the dehydration phenomenon as well.

## 4. Conclusions

In this study, the banana sample was considered a multiphase porous medium, which provided a realistic food material. To investigate the effect of the ripening states of the banana on temperature, vapor, and pressure distributions during dehydration with microwave heating, a computer simulation based on the finite element method was developed. The simulated results correlated well with the experimental results.

The temperature distribution profiles of both unripe and ripe banana samples are non-uniform. The heat generated at the center of the sample rapidly increased with time, while the external regions had lower temperatures, and hot and cold spots were identified. The difference between hot and cold spots was around 70 °C in the case of ripe bananas and 50 °C in case of unripe bananas. The water convective flux peaked at around 11 mm from the center. The vapor pressure peaked at 17 kPa in the case of ripe bananas and 14 kPa in the case of unripe bananas; it occurred at the center in all cases. The moisture, vapor, and pressure distribution profiles showed similar phenomena, which correlated with the temperature distribution profiles. High temperatures caused moisture immigration and the phase transition of water within the sample. At the center, at the highest temperature point, the results showed that the water saturation, vapor mass fraction, and vapor pressure were higher than at the external regions due to the evaporation of the water affecting the water behavior within the sample. The water and vapor fluxes were also correlated with this water behavior. Both unripe and ripe banana samples showed similar patterns to each other, for all profiles. However, they showed slight differences in the levels of values due to their differing initial moisture contents.

Understanding these phenomena is beneficial for the improvement, development, and design of the optimum process for dehydration in bananas with microwave heating. The identification of hot-spot and cold-spot regions can help to avoid the problem of overheated or undercooked areas, which degrade the quality of products and also result in a failure to reach the necessary temperatures to eliminate microorganisms, thereby raising the risk of food deterioration and foodborne illnesses.

**Author Contributions:** Investigation, software, and writing—original draft preparation, W.T.; conceptualization, methodology, and writing—review and editing, K.B.

**Funding:** This research received no external funding.

**Conflicts of Interest:** The authors declare no conflict of interest.

## Nomenclature

| | |
|---|---|
| $E$ | Electric field intensity (V/m) |
| $H$ | Magnetic field intensity (A/m) |
| $\omega$ | Angular frequency (rad/s) |
| $\varepsilon_0$ | Permittivity of free space ($8.854 \times 10^{-12}$ F/m) |
| $j$ | Complex number operator |
| $\varepsilon$ | Complex relative permittivity |
| $\varepsilon'$ | Dielectric constant |
| $\varepsilon''$ | Dielectric loss factor |
| $\mu_r$ | Relative permeability of the food material |
| $\varepsilon_r$ | Relative permittivity |
| $k_0$ | Wave number |

| | |
|---|---|
| $\sigma$ | Electrical conductivity (S/m) |
| $Q_m$ | Heat generation due to microwaves (W/m$^3$) |
| $f$ | Frequency (Hz) |
| $\varepsilon''$ | Dielectric loss factor |
| $\Delta V$ | Representative elementary volume (m$^3$) |
| $\Delta V_g$ | Volume of gas (m$^3$) |
| $\Delta V_w$ | Volume of water (m$^3$) |
| $\Delta V_s$ | Volume of solid (m$^3$) |
| $\varphi$ | Equivalent porosity |
| $S_w$ | Equivalent saturation of water |
| $S_g$ | Equivalent saturation of gas |
| $\rho_v$ | Mass density of vapor (kg/m$^3$) |
| $\rho_a$ | Mass density of air (kg/m$^3$) |
| $\rho_g$ | Mass density of gas (kg/m$^3$) |
| $p_v$ | Partial pressure of vapor (Pa) |
| $p_a$ | Partial pressure of air (Pa) |
| $P$ | Total pressure (Pa) |
| $M_v$ | Molar mass of vapor (kg/mol) |
| $M_a$ | Molar mass of air (kg/mol) |
| $M_g$ | Molar mass of gas (kg/mol) |
| $\Delta m_v$ | Mass of vapor in a representative elementary volume (kg) |
| $\Delta m_a$ | Mass of air in a representative elementary volume (kg) |
| $R$ | Universal gas constant (J/mol K) |
| $T$ | Temperature (K) |
| $c_v$ | Mass concentration of vapor (kg/m$^3$) |
| $c_a$ | Mass concentration of air (kg/m$^3$) |
| $c_w$ | Mass concentration of liquid water (kg/m$^3$) |
| $\Delta m_w$ | Mass of liquid water in a representative elementary volume (kg) |
| $\rho_w$ | Density of water (kg/m$^3$) |
| $k_w$ | Intrinsic permeability of water (m$^2$) |
| $k_{r,w}$ | Relative permeability of water |
| $\mu_w$ | Viscosity of water (Pa s) |
| $D_c$ | Capillary diffusivity (m$^2$/s) |
| $R_{evap}$ | Evaporation rate of liquid water to water vapor (kg/m$^2$ s) |
| $\omega_v$ | Mass fraction of vapor |
| $k_g$ | Intrinsic permeability of gas (m$^2$) |
| $k_{r,g}$ | Relative permeability of gas |
| $\mu_g$ | Viscosity of gas (Pa s) |
| $D_{eff,g}$ | Binary diffusivity of vapor and air (m$^2$/s) |
| $\omega_a$ | Mass fraction of air |
| $\vec{n}_w$ | Water flux (kg/m$^2$ s) |
| $\vec{n}_g$ | Gas flux (kg/m$^2$ s) |
| $h_g$ | Enthalpy of gas (J) |
| $h_w$ | Enthalpy of water (J) |
| $h_{fg}$ | Latent heat of evaporation (J/kg) |
| $\rho_{eff}$ | Effective density (kg/m$^3$) |
| $c_{peff}$ | Effective specific heat (J/kg K) |
| $k_{eff}$ | Effective thermal conductivity (W/m K) |
| $\rho_s$ | Solid density (kg/m$^3$) |
| $c_{pg}$ | Specific heat capacity of gas (J/kg K) |
| $c_{pw}$ | Specific heat capacity of water (J/kg K) |
| $c_{ps}$ | Specific heat capacity of solid (J/kg K) |
| $k_{th,g}$ | Thermal conductivity of gas (W/m K) |
| $k_{th,w}$ | Thermal conductivity of water (W/m K) |

| | |
|---|---|
| $k_{th,s}$ | Thermal conductivity of solid (W/m K) |
| $m_g$ | Mass fraction of gas (kg) |
| $m_w$ | Mass fraction of water (kg) |
| $m_s$ | Mass fraction of solid (kg) |
| $K_{evap}$ | Evaporation constant (1/s) |
| $p_{v,eq}$ | Equilibrium vapor pressure (Pa) |
| $\vec{n}_{w,s}$ | Total water flux at the surface (kg/m$^2$ s) |
| $\vec{n}_{v,s}$ | Total vapor flux at the surface (kg/m$^2$ s) |
| $h_{mv}$ | Mass transfer coefficient (m/s) |
| $p_{v,air}$ | Vapor pressure of ambient air (Pa) |
| $P_{amb}$ | Ambient pressure (Pa) |
| $h_T$ | Heat transfer coefficient (W/m$^2$K) |
| $T_{air}$ | Drying air temperature (K) |
| $C_{p,v}$ | Specific heat of vapor at constant volume (J/kg K) |
| $d_p$ | Penetration depth in the material (mm) |
| $\lambda$ | Wavelength (m) |
| $\tan \delta$ | Dielectric loss tangent |

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
