# Peer review of "Investigation of Heat and Moisture Transport in Bananas during Microwave Heating Process"

_processes, doi:10.3390/pr7080545_

Round 1
Reviewer 1 Report
General comments
The paper deals with a CFD model on banana drying. The topic is interesting and the methodology is correct. The analysis is highly detailed and of great interest. Some advices are provided on how to improve literature review and presentation. The English Language should be corrected recurring to a mother tongue sicentific editor;
Specific comments
- Correct the English Language recurring to a mother tongue scientific editor;
- Insert a nomenclature reporting the symbols contained in the equations
- line 205 page 6, when you introduce the evaporation constant say please how this has been determined. You could have determined this also experimentally , see for example:
Bartocci, P., Tschentscher, R., Stensrød, R.E., Barbanera, M., Fantozzi, F., Kinetic analysis of digestate slow pyrolysis with the application of the master-plots method and independent parallel reactions scheme (2019) Molecules, 24 (9), art. no. 1657 - check the quality of th figures. The have low definition. check that x axis and y axis have Always the unit of measure. If the quantity presented in dimensionless, please insert the following unit of measure: (-). - insert more quantitative data in the conclusions.
Reviewer 2 Report
This paper deals with numerical analysis of heat and moisture transport of during microwave heating of banana for dehydration to enhance their preservation. COMSOL multi-physics software was used to simulate the microwave irradiation effects on banana with different ripening stage. The simulation results exhibited good agreement with the microwave heating experiments. Only minor revisions are required as following comments.
(1) Please indicate the size of multimode microwave oven.
(2) Please indicate the changes in the dielectric property of banana during microwave heating. It should change depending on the temperature, moisture content and some color formation as indicated in Fig. 11.
(3) Please indicate the overall electromagnetic field distribution.
(4) Please indicate the exact moisture content of banana used in the microwave heating experiment. This value should change very easily.
(5) How the homogeneity of microwave distribution was achieved? Turn table or stirrer fan? This should be critical for formation of hot spot.
(6) Description of microwave apparatus should be indicated (model, power). How the microwave power was controlled?
(7) The only cut banana was used in this study. But this should be different from the real microwave heating process of banana. How about other geometry of banana?
Round 2
Reviewer 1 Report
All the required changes have been performed. Paper can be accepted.